# DMID: Dynamic Mask Attention for High-Fidelity Identity Preservation under Limited Data

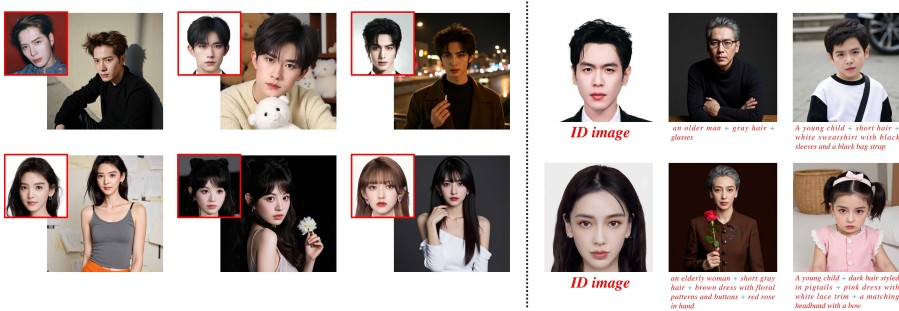

Figure 1: DMID ensures high-fidelity identity preservation while preserving textual semantics. On the left are examples of some famous figures, and on the right are examples of age editing.

## Abstract

We present Dynamic Mask Attention for High-Fidelity Identity Preservation under Limited Data (DMID), which aims to precisely reconstruct fine-grained identity features under scarce data conditions while alleviating conflicts between textual and conditional semantics. At its core, DMID employs a Variational Autoencoder (VAE) for meticulous identity encoding and introduces a **Dynamic Attention Mask mechanism**, coupled with **Distribution Consistency Loss** and **Identity Mask Loss**, ensuring identity fidelity while mitigating semantic conflicts. To further reduce annotation and training costs, we have designed an efficient data construction pipeline. Furthermore, our method enables the dynamic adjustment of the **AttnMask strength factor** during inference, ensuring precise modifications and fine-grained control over identity features and semantics across various scenarios. The training process is divided into three stages: (1) identity embedding stage, (2) dynamic attention mask learning stage, and (3) Diffusion-DPO post-training stage. Evaluated on our newly constructed ID Benchmark, DMID achieves state-of-the-art performance in both identity consistency and textual semantics, demonstrating its strong competitiveness in data-limited scenarios. Among them, the parameter count of AttnMaskNet is only approximately 1% of that of Flux.1-dev.

## 1 Introduction

In text-to-image (T2I) generationEsser et al. (2024); Rombach et al. (2022); Peebles & Xie (2023); Chen et al. (2023); Betker et al. (2023), Transformer-based diffusion models such as Flux Labs (2024) can produce high-fidelity images with complex semantics. However, maintaining consistent human identity (ID) across diverse generated scenes remains challenging, as existing techniques struggle to balance identity fidelity and textual semantics.

Early personalization methods, such as LoRA Devalal & Karthikeyan (2018) and DreamBooth Ruiz et al. (2023), rely on tens of target identity images for fine-tuning, which poses challenges to the robustness of identity consistency tasks.Recent approachesLi et al. (2024); Peng et al. (2024); Xiao

et al. (2025); Yuan et al. (2023); Han et al. (2024) including IP-Adapter Ye et al. (2023) and InstantID Wang et al. (2024b) significantly improve ID consistency by extracting identity features through auxiliary facial encoders and injecting them into generative models. While enabling personalized generation with single/few reference images, they suffer from inherent information compression during facial encoding that causes identity detail loss. Moreover, the introduced conditioning tokens frequently conflict with textual semantic tokens, reducing prompt responsiveness.

To address semantic degradation, Pulid-Flux Guo et al. (2024) introduces semantic alignment loss and layout alignment loss, while InfiniteYou Jiang et al. (2025) designs InfuseNet for improved fusion of facial encodings with foundation models. Despite these advances, effective alignment between external facial features and the foundation model's latent space demands 100K to 1M high-quality training samples. The substantial data and computational costs severely limit reproducibility and practical deployment, highlighting the need for robust low-cost solutions.

OmniControl Tan et al. (2024) adopts a distinct approach: it directly encodes an additional condition image into the feature space of the foundation model using a pre-trained VAE encoder Kingma et al. (2013), thus unifying the encodings of condition image and noise image in a shared space. Similar structures were also used in earlier works on ReferenceNet Hu (2024); Xu et al. (2024b); Tian et al. (2024); Chang et al. (2023); Xu et al. (2024a); Choi et al. (2024); Wang et al. (2024c); Huang et al. (2024); Gu et al. (2024); Zhang et al. (2023). This strategy, which ensures high consistency between the encodings of conditional input and noise, significantly enhances the training efficiency and consistency of the details. Although the method demonstrates great potential for various tasks, such high consistency leads to the issue of reference image replication. Additionally, the number of conditional tokens is much greater than in earlier work such as IP-Adapter Ye et al. (2023), which exacerbates conflicts between textual semantics and image semantics.

To preserve high facial identity consistency while minimizing semantic interference, we propose a high-fidelity, identity-consistent diffusion model driven by dynamic attention masks. Our framework employs a multistage learning approach incorporating a dynamic attention masking mechanism guided by two losses: (1) **Distribution Consistency Loss**, which aligns conditional and textual feature distributions; and (2) **Identity Mask Loss**, which directly optimizes facial similarity between generated and target ID image.

This mechanism compresses identity information into a facial-specific subspace and dynamically masks conflict regions between tokens, achieving robust identity preservation with minimal training data (approximately 40,000 high-quality image pairs). We also develop a specialized data acquisition pipeline to resolve semantic conflicts and source image replication. Our principal contributions:

**Consistency-driven dynamic attention masking**: A novel masking scheme utilizing distribution consistency loss and identity mask loss, significantly enhancing face similarity while maintaining textual semantic consistency. We introduce the AttnMask strength factor, which dynamically adjusts attention levels during inference, enabling a precise balance between identity preservation and semantic consistency, thus allowing fine-grained control and preventing semantic distortion due to over-attention.

**Efficient multi-stage training framework**: A three-phase approach leveraging unified VAE encoding:

*Stage 1 (Identity Embedding)*: Learns identity embeddings from condition image.

*Stage 2 (Dynamic Mask Optimization)*: Jointly trains AttnMaskNet using both consistency losses.

*Stage 3 (Preference Alignment)*: We adopt Diffusion-DPO Wallace et al. (2024) training to enhance the stability of model generation.

**ID consistency benchmark**: We establish and open-source a standardized evaluation benchmark spanning diverse demographics (race, gender, age) with standardized protocols.

## 2 RELATED WORK

### 2.1 ID CONSISTENT GENERATION

To maintain specified identity (ID) in text-to-image generation, mainstream approaches focus on feature injection and model optimization. Feature injection adapters (e.g., IP-Adapter Ye et al. (2023), InstantID Wang et al. (2024b)) utilize pre-trained encoders to extract facial features, injecting them into foundation models (e.g., SDXL Podell et al. (2023)) via lightweight plugins (e.g., cross-attention). However, feature compression causes detail loss, while additional conditioning tokens frequently conflict with text tokens. Loss optimization methods (e.g., PuLID Guo et al. (2024)) explicitly incorporate ID loss based on face recognition models (e.g., ArcFace Deng et al. (2019)), directly optimizing generated face similarity to achieve quantifiable metric improvements. Architectural innovations explore new pathways: FlashFace Zhang et al. (2024) trains end-to-end isomorphic UNet encoders for fine-grained fusion, while ACE++ Mao et al. (2025) unifies conditional and noise image in DiT architecture by mapping to VAE latent space Kingma et al. (2013). Although enhancing detail consistency, this tends to cause reference image over-replication and exacerbates semantic conflicts due to excessive conditional tokens.

### 2.2 HUMAN PREFERENCE ALIGNMENT IN DIFFUSION MODELS

Inspired by Reinforcement Learning from Human Feedback (RLHF) paradigms in large language models (LLMs), alignment research for diffusion models has expanded rapidly. DRAFT Clark et al. (2023) and AlignProp Prabhudesai et al. (2023) integrate reward model gradients directly into training. Diffusion-DPO Wallace et al. (2024) adopts an offline approach using preference pairs, eliminating reward models as a simpler RLHF alternative. Concurrently, Flow-GRPO Liu et al. (2025) and DanceGRPO Xue et al. (2025), MixGRPOLi et al. (2025) embed online reinforcement learning within flow-matching frameworks, demonstrating significant gains in human preference tasks.

## 3 METHODS

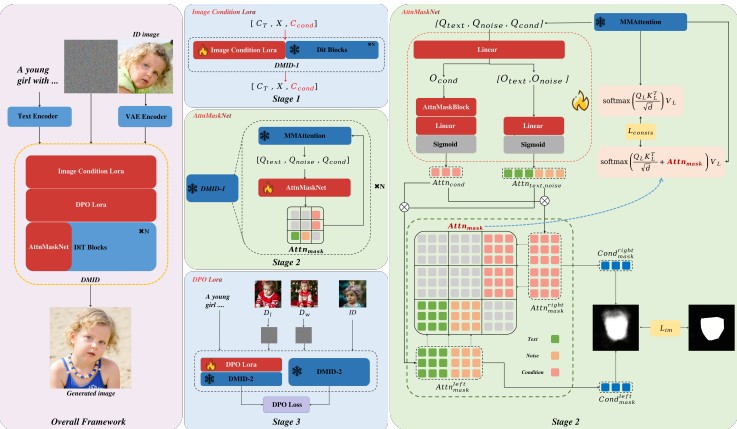

Figure 2: **Overall training framework of DMID**, consisting of three core stages: 1) The identity embedding stage; 2) The dynamic attention mask learning stage; 3) The Diffusion-DPO post-training stage. The AttnMaskNetBlock primarily comprises residual blocks and convolutional layers.

### 3.1 ENSEMBLE-BASED CONDITIONAL CONTROL LoRA

To achieve ID conditioned image embedding, we adopt the framework of Ominicontrol Tan et al. (2024) for the first-stage training. Specifically, we employ Rectified Flows Liu et al. (2022) as the forward sampling process and use Conditional Flow Matching Lipman et al. (2022) as the optimiza-

tion objective, with the loss function defined as:

$$\mathcal{L}_{\text{CFM}} = \mathbb{E}_{t,p_t(z|\epsilon),p(\epsilon)} \left\| v_\theta(z,t) - u_t(z|\epsilon) \right\|_2^2 \quad, \tag{1}$$

where $\epsilon \sim \mathcal{N}(0, I)$ and $v_\theta$ is parameterized by neural network weights. The detailed network structure is illustrated in Stage 1 of Figure 2. First, the condition image is encoded via a VAE to obtain condition image tokens $C_{\text{cond}} \in \mathbb{R}^{K \times d}$. These tokens are then integrated with text tokens $C_T \in \mathbb{R}^{M \times d}$ and image tokens $X \in \mathbb{R}^{N \times d}$ into a unified space $L = [C_T, X, C_{\text{cond}}]$ to allow interactions between multiple conditions. To further enhance flexibility, Ominicontrol proposes a method for manually controlling the strength of condition image. It constructs an attention mask matrix $Attn(\alpha)$ using a given strength factor $\alpha$ to adjust attention weights between condition image tokens and other tokens:

$$\text{MMAttention}(L) = \sigma \left( \frac{Q_L K_L^\top}{\sqrt{d}} + Attn(\alpha) \right) V_L \quad, \tag{2}$$

where $\sigma(\cdot) \triangleq \text{softmax}(\cdot)$ denotes the softmax function, and $Attn(\alpha)$ is a structured masking matrix that selectively modifies attention weights for the condition image tokens. Specifically, $Attn(\alpha)$ is defined as:

$$Attn(\alpha) = \begin{bmatrix} 0_{M \times M} & 0_{M \times N} & \alpha_{M \times K} \\ 0_{N \times M} & 0_{N \times N} & \alpha_{N \times K} \\ \alpha_{K \times M} & \alpha_{K \times N} & 0_{K \times K} \end{bmatrix} \quad. \tag{3}$$

While the strength factor setting effectively controls the condition image effect, it suffers from several drawbacks. First, the strength factor operates globally across the entire condition image, leading to re-weighting of irrelevant regions that should be suppressed. In the face identity preservation task, the current method assigns equal weight to the face and the background. In fact, attention should be focused on the face area to accurately maintain identity consistency. Second, when the strength factor is set too large, it will seriously damage the original attention distribution and lead to the loss of text semantic information.

To address these issues, we design a dynamic Attention Mask learning scheme. The model learns an adaptive Attention Mask to enhance attention to the main subject while ensuring no significant loss of textual semantics.

### 3.2 Dynamic Attention Mask Learning

As shown in Stage 2 of Figure 2, we take the query matrix from the attention module as input to AttnMaskNet and finally obtain an attention mask that acts on the condition image region. Its function is to improve face similarity while keeping the text and image as aligned as possible. As mentioned earlier, over-enhancement of some regions will disrupt the overall distribution, leading to text-image misalignment. Therefore, our main idea is to help the model suppress redundant attention regions and enhance key regions, which can maximize the attention enhancement while mitigating the loss of semantic information. We have designed corresponding components from the aspects of network structure and loss functions.

#### 3.2.1 AttnMaskNet

As shown in Figure 2, the detailed workflow of AttnMaskNet is illustrated. Although our focus is on modifying the attention of condition image, we need to consider the original attention relationships between text, noise, and condition image. To preserve the inherent attention relationships among different conditional tokens in the model, we concatenate the query matrices of text, noise, and condition image from the attention module as the input to the network $[Q_{\text{text}}, Q_{\text{noise}}, Q_{\text{cond}}] \in \mathbb{R}^{L_1 \times d_1}$ where $L_1 = M + N + K$.

The projected output is divided into the text-noise part $Output_1 = [O_{\text{text}}, O_{\text{noise}}] \in \mathbb{R}^{L_2 \times d_2}$ (where $L_2 = M + N$) and the condition image part $Output_2 = O_{\text{cond}} \in \mathbb{R}^{K \times d_2}$. We simply project the text-noise part further to retain their original attention information, ultimately obtaining $Attn_{\text{text,noise}} \in \{ \mathbf{x} \in \mathbb{R}^{L_2} \mid 0 \leq x_i \leq 1, \forall i = 1, \ldots, L_2 \}$, where each element represents the strength of attention of each token in text and noise towards the condition image.

For the condition image part, we aim to enable it to focus precisely and effectively on key regions while masking non-critical ones. Attention modules can effectively capture global information, but

are relatively weak in local spatial information. To obtain more refined spatial features, $Output_2$ is reshaped into $\mathbb{R}^{d_2 \times \sqrt{K} \times \sqrt{K}}$ and fed into a residual network. Finally, the output of the residual network is reshaped and projected to obtain $Attn_{\text{cond}} \in \left\{ \mathbf{x} \in \mathbb{R}^K \mid 0 \leq x_i \leq 1, \forall i = 1, \ldots, K \right\}$.

Meanwhile, to ensure the attention mask can effectively enhance or suppress attention weights, we map the $Attn_{cond}$ of each condition to the interval $[-\beta, \beta]$:

$$Attn'_{cond} = 2\beta \cdot Attn_{cond} - \beta \ , \tag{4}$$

where $Attn_{cond}$ denotes the original matrix with elements in $[0, 1]$, and $\beta$ is the strength factor controlling the range of the transformed values ($\beta$ is set to 1 by default unless explicitly stated otherwise.). The same transformation yields $Attn'_{text,noise} \in \mathbb{R}^{L_2}$.

As shown in Figure 2, we compute the regions of $Attn_{\text{mask}}$ corresponding to the condition image, i.e., the upper-right region $Attn_{\text{mask}}^{\text{right}} = Attn_{\text{text,noise}}^{\top} Attn'_{\text{cond}}$, $Attn_{\text{mask}}^{\text{right}} \in \mathbb{R}^{L_2 \times K}$ and the lower-left region $Attn_{\text{mask}}^{\text{left}} = Attn_{\text{cond}}^{\top} Attn'_{\text{text,noise}}$, $Attn_{\text{mask}}^{\text{left}} \in \mathbb{R}^{K \times L_2}$. The resulting attention mask is:

$$Attn_{\text{mask}} = \begin{bmatrix} 0_{(M+N) \times (M+N)} & Attn_{\text{mask}}^{\text{right}} \\ Attn_{\text{mask}}^{\text{left}} & 0_{K \times K} \end{bmatrix} \ , \tag{5}$$

The obtained $Attn_{\text{mask}}$ acts on the attention map as a weight matrix, as shown in Equation 5. Here, $Attn_{\text{mask}}^{\text{right}}$ and $Attn_{\text{mask}}^{\text{left}}$ preserve the original attention relationships between text, noise, and the condition image while learning the information that needs to be enhanced or suppressed. This avoids disrupting the original self-attention distribution of the model and helps correct attention effectively. As a branch network, AttnMaskNet is incorporated into the attention module at every layer of the model. To guide AttnMaskNet toward our desired learning direction, we further design an identity mask Loss and a distribution consistency loss.

### 3.2.2 IDENTITY MASK LOSS

To help the model effectively learn key and non-key regions, we introduce a face mask as prior information to constrain the learning process of $Attn_{\text{mask}}$. As shown in Stage 2 of Figure 2, we first average $Attn_{\text{mask}}^{\text{right}}$ and $Attn_{\text{mask}}^{\text{left}}$ over the text and noise dimensions, yielding the condition image vectors $Cond_{\text{mask}}^{\text{right}}$ and $Cond_{\text{mask}}^{\text{left}}$ with dimension $\mathbb{R}^K$. These vectors are then resized to match the spatial size of the face mask, and the pixel-wise error between them is computed using binary cross-entropy Wang et al. (2024d):

$$\mathcal{L}_{\text{im}} = -\frac{1}{K} \sum_{i=1}^{\sqrt{K}} \sum_{j=1}^{\sqrt{K}} \hat{f}_{i,j} \log(\hat{a}_{i,j}) + (1 - \hat{f}_{i,j}) \log(1 - \hat{a}_{i,j}) \ , \tag{6}$$

where $\hat{f}_{i,j}$ denotes the pixel value of the face mask, $\hat{a}_{i,j}$ denotes the pixel value corresponding to the reshaped condition image vector, and $\sqrt{K}$ is the spatial dimension size.

### 3.2.3 DISTRIBUTION CONSISTENCY LOSS

ID consistency and text-image alignment are inherently conflicting. Essentially, when the model over-focuses on the condition image, the original attention distribution is disrupted, leading to loss of semantic information. Experimental observations show that increasing the scale of the condition image improves face similarity but weakens textual semantics. Expanding the softmax operation in Equation 2, the probability value for the text region can be expressed as:

$$M_{ik} = \frac{e^{x_{ik}}}{\sum_{j=1}^{M} e^{x_{ij}} + \sum_{j=M+1}^{M+N} e^{x_{ij}} + \sum_{j=M+N+1}^{L_1} e^{x_{ij}+a_{ij}}} \ , \tag{7}$$

where $0 < i < L_1, 0 < k < M$. Let $S_c = \sum_{j=M+N+1}^{L_1} e^{x_{ij}+a_{ij}}$, and $a_{ij}$ denotes an element in $Attn_{\text{mask}}$. When $a_{ij} \rightarrow -\infty$, the expression degenerates to the standard text-to-image formulation.

It can be seen from the above formula that an increase in $S_c$ leads to a decrease in $M_{ik}(0 < k < M)$. To reduce the weakening of textual semantics by adjusting $a_{ij}$, an effective method is to keep $S_c$ unchanged. Therefore, we introduce a distribution consistency loss:

$$\mathcal{L}_{\text{consis}} = \frac{1}{L_1} \sum_{i=1}^{L_1} \left\| \sum_{k=M+N+1}^{L_1} M_{ik} - \sum_{k=M+N+1}^{L_1} M_{ik}^* \right\| , \tag{8}$$

where $M_{ik}^*$ denotes the probability value when $a_{ij} = 0$, which is also the target value of the loss function. This regularizes the sum of probability values in the condition image region, indirectly preserving the probability of the text region to alleviate semantic conflicts.

The final optimization objective:

$$\mathcal{L}_{\text{stage2}} = \mathcal{L}_{\text{CFM}} + \lambda \left( \mathcal{L}_{\text{im}} + \mathcal{L}_{\text{consis}} \right) , \tag{9}$$

where $\lambda$ is a hyperparameter ($0 < \lambda < 1$) that balances the weights of identity mask loss and distribution consistency loss.

### 3.3 DIFFUSION-DPO

To address the instability issue caused by the absence of perceptual loss (e.g. PuLID) in VAE-based image encoding during training, where the similarity of generated results fluctuates significantly under the same image conditions, this paper employs Diffusion-DPO for post-training refinement. As shown in Figure 2, Diffusion-DPO requires offline construction of paired preference data. To this end, we design a comprehensive scoring model:

$$R = R_{\text{CosSim}} + \mu R_{\text{CLIP}} , \tag{10}$$

where $R_{\text{CosSim}}$ denotes the facial similarity metric, $R_{\text{CLIP}}$ represents the semantic consistency metric from CLIP model Radford et al. (2021), and $\mu$ balances their contributions. During the offline phase, we generate multiple outputs for the same input using different random seeds, compute their scores, and select the highest-scoring output as the winning sample $D_w$ and the lowest-scoring one as the losing sample $D_l$ to form paired preference data for Diffusion-DPO optimization.

## 4 EXPERIMENT

### 4.1 EXPERIMENTAL SETUP

#### 4.1.1 IMPLEMENTATION DETAILS

We implement DMID using PyTorch and HuggingFace Diffusers. The DiT base model is FLUX.1-dev. All experiments were conducted on 2×NVIDIA H100 GPUs. The training batch size is 1 with gradient accumulation steps of 2, and target image size is 768×768. For Stage 1 training, we use the Prodigy optimizerMishchenko & Defazio (2024) with safe warmup and bias correction, setting weight decay to 0.01 for 30,000 iterations. In Stage 2, we train for 2,000 iterations: the first 1,500 steps use weight coefficient $\lambda = 1$, and the last 500 steps use $\lambda = 0.1$. For Stage 3 DPO training, we fix $W = 5000$, $\mu = 4$ and run 2,500 iterations with AdamW optimizerLoshchilov & Hutter (2017) (learning rate=0.0001, $\beta_1 = 0.9$, $\beta_2 = 0.999$).

#### 4.1.2 TRAINING DATASET

Through our data construction pipeline, we obtain around 20,000 image pairs—totaling 40,000 individual images—covering 12,000 unique IDs. For video generation, we use Wan2.1Wan et al. (2025) for batch image-to-video conversion, sampling frame pairs at 20fps. For image augmentation, ACE++ generates diverse pairs through different mask inputs. Low-similarity data undergoes face swappingZhou et al. (2022) to enhance ID consistency. Based on the first-stage training, we develop a local face- and head-swapping ID-inpainting model and, by leveraging an external aesthetic LoRA, produce high-quality data pairs in batches. The data filtering pipeline includes: (1) resolution filtering (768px), (2) aesthetic scoring $> 5.2$ using aesthetic-predictor-v2-5, (3) face quality filtering (removing occluded, multi-person, or extreme-pose image via face detection and landmark models), and (4) similarity filtering with ArcFace threshold $> 0.8$. All images are annotated using Qwen2-VL-7BWang et al. (2024a) for person and background descriptions.

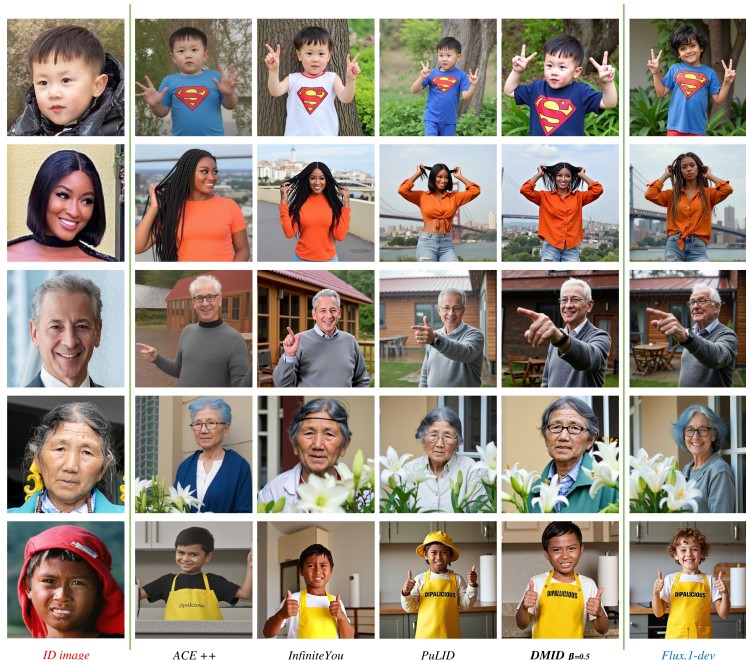

ID image     ACE ++     InfiniteYou     PuLID     **DMID** β=0.5     Flux.1-dev

Figure 3: Qualitative comparison. A qualitative comparison of DMID with the latest baseline methods including PuLID-FLUX, InfiniteYou, and ACE++. To demonstrate that DMID can maintain good image-text consistency, we also use FLUX.1-dev to generate examples without an ID image. More example demonstrations can be found in the supplementary materials.

### 4.1.3 BASELINES AND EVALUATION

Since DMID is trained on Flux.1-dev with VAE-encoded image conditions, we compare against three state-of-the-art DiT-based methods: PuLID-FLUX, InfiniteYou, and ACE++ (Portrait LoRA). For evaluation, existing works use private small-scale ID datasets (e.g., 170 IDs in PuLID, 15 IDs in InfiniteYou). To enable fair comparison, we created a new ID consisitency benchmark consisting of 504 unique IDs covering three age groups (elderly, young, children) and four races (Caucasian, African, Asian, South Asian) from LAION-human Ju et al. (2023); Schuhmann et al. (2022). Each test case includes a pair of images with different IDs, and all images are annotated using Qwen2-VL-7B Wang et al. (2024a).

### 4.2 COMPARISON WITH PRIOR METHODS

### 4.2.1 QUALITATIVE COMPARISON

As shown in Figure 3, our qualitative comparison reveals distinct performance characteristics. Methods using facial encoders (PuLID and InfiniteYou) maintain reasonable ID consistency but fail to capture fine-grained facial features like wrinkles and spots. While preserving basic semantic structures, their outputs exhibit noticeable artificial generation artifacts. In contrast, VAE-encoded approaches (ACE++

Table 1: Quantitative comparison results

| Method | CosSim ↑ | CLIPScore ↑ |
|---|---|---|
| Ace++ | 0.5095 | 30.4532 |
| InfiniteYou | 0.6724 | 31.4062 |
| PuLID | 0.7051 | 32.0730 |
| **DMID**$_{\beta=0.5}$ | **0.7672** | **32.4694** |

and our DMID) produce more photorealistic results with accurate reproduction of subtle facial details, as demonstrated by the faithful rendering of elderly subjects' spots and wrinkles in the fourth example column. Crucially, DMID achieves superior image-text alignment, maintaining not only semantic structures but also precise pose preservation and textual element fidelity across all examples. These visual comparisons confirm DMID's state-of-the-art performance in terms of identity consistency and textual semantics.

### 4.2.2 QUANTITATIVE COMPARISON

Quantitative results are presented in Table 1. We employ CLIPScore to measure image-text semantic alignment, which evaluates the model's ability to follow input prompts. Cosine similarity (CosSim) is used to assess facial identity consistency. The results demonstrate that DMID achieves state-of-the-art performance in both identity consistency and textual semantics.

Table 2: Performance Comparison of Models

| Experiment | Model | CosSim ↑ | CLIPScore ↑ |
|---|---|---|---|
| | DMID-1 | 0.4906 | 32.6607 |
| **Exp 1** | DMID-1$_{\alpha=1.2}$ | 0.6288 | 31.8678 |
| | DMID-1$_{\alpha=1.5}$ | 0.7455 | 30.3416 |
| | DMID-2 $w/o\ L_{\text{im}}$ | 0.4890 | 32.7084 |
| **Exp 2** | DMID-2 $w/o\ L_{\text{consis}}$ | 0.7595 | 31.5468 |
| | DMID-2 | 0.7021 | 32.2621 |
| | DMID | 0.7803 | 32.1851 |
| | DMID$_{\beta=0.75}$ | 0.7712 | 32.3485 |
| **Exp 3** | DMID$_{\beta=0.6}$ | 0.7722 | 32.4371 |
| | DMID$_{\beta=0.5}$ | 0.7672 | 32.4694 |
| | DMID$_{\beta=0.4}$ | 0.7611 | 32.5147 |

## 4.3 ABLATION STUDIES

### 4.3.1 IMPACT OF STRENGTH FACTOR $\alpha$

We conduct ablation experiments to validate the effectiveness of different components in our proposed method and the multi-stage training strategy. First, we present a preliminary experiment to illustrate the motivation behind our approach. We set the strength factor $\alpha$ using the methods described in Equations 2 and 3, and evaluate performance on our proposed ID benchmark.

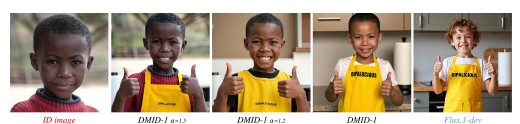

Figure 4: Demonstration of strength factor $\alpha$ testing.

Figure 4 shows selected test cases. It can be observed that DMID-1 preserves the original semantic structure, including complete text information. As the strength factor $\alpha$ increases, the similarity between the generated face and the ID image face improves significantly; however, the original background, clothing, and text undergo noticeable changes. The quantitative results in Exp 1 of Table 2 are consistent with these visual observations. These findings motivate our dynamic mask learning scheme.

### 4.3.2 ABLATION STUDY ON DYNAMIC ATTENTION MASK

To validate the effectiveness of the proposed dynamic masking scheme, we conduct an ablation study on Stage 2 components. This stage incorporates two loss functions ($\mathcal{L}_{\text{consis}}$ and $\mathcal{L}_{\text{im}}$), whose individual contributions are examined in Figure 5 and Exp 2 of Table 2. Figure 5 visualizes generated samples with corresponding identity-focused attention heatmaps, revealing three key observations:

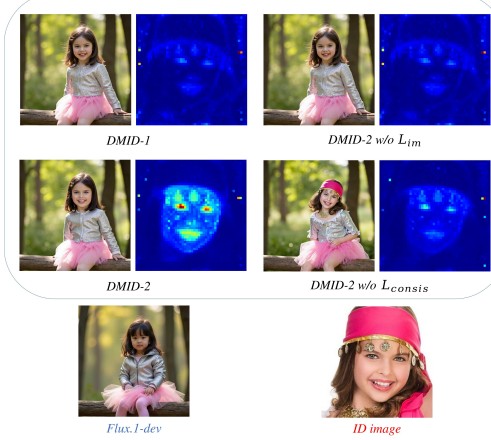

Figure 5: Ablation study on dynamic attention mask learning.

**DMID-2 w/o $\mathcal{L}_{\text{im}}$**: As shown in Table 2, the model fails to effectively adjust its attention distribution, resulting in negligible visual changes compared to the baseline (DMID-1). **DMID-2 w/o $\mathcal{L}_{\text{consis}}$**: Excessive attention on facial regions induces copy-paste artifacts and mouth distortions. Quantitative metrics align with visual observations: facial similarity increases while CLIP scores significantly drop.

**DMID-2**: When $\mathcal{L}_{\text{consis}}$ and $\mathcal{L}_{\text{im}}$ are optimized together, the model effectively focuses on key facial regions, improving facial similarity while reducing the loss of semantic information. Notably, DMID-2 alone already achieves performance on par with the current state-of-the-art PuLID-Flux.

### 4.3.3 MULTI-STAGE TRAINING STRATEGY

Table 2 and Figure 6 demonstrate the effectiveness of our multi-stage training strategy. As shown in Table 2, facial similarity metrics significantly improve with minimal semantic information loss. This phenomenon is also reflected in visual results: Figure 6 shows that as training progresses, the model captures increasingly fine-grained facial features such as wrinkles and spots. Therefore, we conclude that the proposed multi-stage training strategy effectively enhances face similarity while reducing semantic loss.

### 4.3.4 AttnMask strength factor $\beta$

During the experiments, we observed a key phenomenon: there exists a saturation threshold for the model's effective attention to identity features. Beyond this threshold, increasing the attention weights not only fails to alter the visual appearance of facial regions but also exacerbates the loss of semantic information. Extensive test results demonstrate that the facial similarity (CosSim) of most samples reaches a saturation threshold between 0.7 and 0.8; further improving this metric only yields redundant gains, while the semantic consistency metric (CLIPScore) continues to decline. Therefore, we introduce the design of an attention mask scaling channel.

Based on this phenomenon, we achieve a refined trade-off between the two consistencies by adjusting the scaling factor $\beta$. The core of this method lies in strategically sacrificing redundant identity attention to specifically enhance semantic control capability. As illustrated in Figure 7, regulating $\beta$ enables: 1) Fine-grained semantic control (decorative attributes such as glasses, hats, and expressions); 2) Complete preservation of identity details (including biological features like tiny spots); 3) Avoidance of semantic distortion caused by over-attention.

The quantitative results in Exp 3 of Table 2 confirm that this method can further improve semantic alignment by sacrificing redundant similarity while maintaining the integrity of identity details. This indicates that the $\beta$-based attention scaling channel effectively resolves the consistency conflict.

Figure 6: Ablation study on multi-stage training strategy.

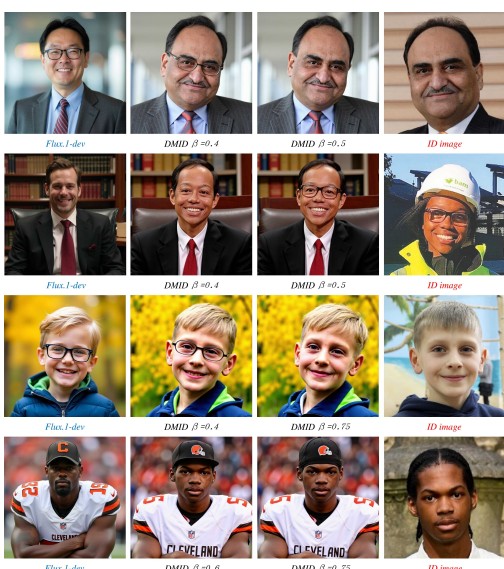

Figure 7: **Strength factor $\beta$ ablation.**

## 5 Conclusion

This paper presents DMID, a high ID-consistency approach for limited-data scenarios. By integrating dynamic attention masking with a joint loss enforcing both identity and distribution consistency, and employing a three-stage progressive training strategy, DMID significantly improves identity fidelity and detail reconstruction while mitigating the conflict between textual semantics and condition image. The AttnMask strength factor design allows more refined editing. On the data side, we develop an automated, high-quality pipeline that constructs training pairs with preserved identity consistency under small-scale data and release the ID Consistency Benchmark, a dataset covering 504 distinct identities. Extensive experiments show DMID outperforms existing methods in both identity consistency and facial detail recovery. Future work can adopt DMID's training paradigm for identity consistency in specific scenarios with minimal data and reduced cost.

## 6 REPRODUCIBILITY STATEMENT

The specific implementation code of the proposed method will be open-sourced, and the anonymous link will be provided in the comments. The data construction pipeline can be referred to in the appendix section.

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

# A APPENDIX

## A.1 THE USE OF LARGE LANGUAGE MODELS

This paper only uses large language models for English translation and English grammar correction and polishing.

## A.2 EXTENDED EXPERIMENT

There is an inherent conflict between facial similarity and textual semantic alignment. Based on this premise, we propose DMID. In the previous section, we demonstrated the strong identity-preserving capability and editability of DMID in real-world scenarios. To further showcase the effectiveness of our proposed method, we will provide additional experimental results.

## A.3 THE USABILITY OF THE METHOD IN OTHER TASKS

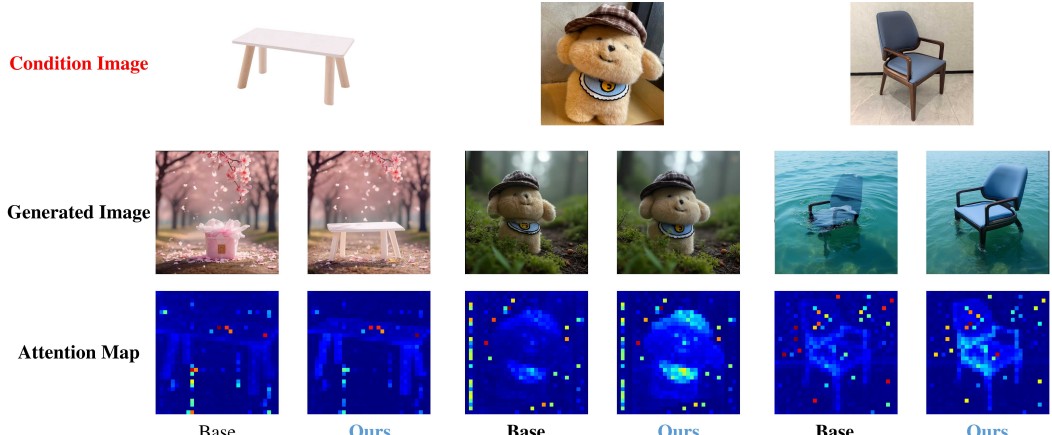

Figure 8: The Usability of the Method in Other Tasks

To verify that the AttnMaskNet scheme is not restricted exclusively to identity consistency preservation tasks, we adopt the open-source subject consistency preservation model *subject_512* from Ominicontrol as the base model—corresponding to the stage-1 model described in our methodology. For the training process of AttnMaskNet, we utilize Subject200K, the official training dataset associated with *subject_512*. The experimental results are presented in the Figure 8, which includes both the generated output images and self-attention maps of the base model (*subject_512*) and our AttnMaskNet-augmented model. As observed from these results, AttnMaskNet retains its effectiveness when applied to subject consistency tasks.

### A.3.1 DMID-INPAINTING

We reproduce the first stage of DMID on our self-constructed dataset and further propose a region-controllable, high-fidelity local identity-preserving model, DMID-Inpainting. DMID-Inpainting not

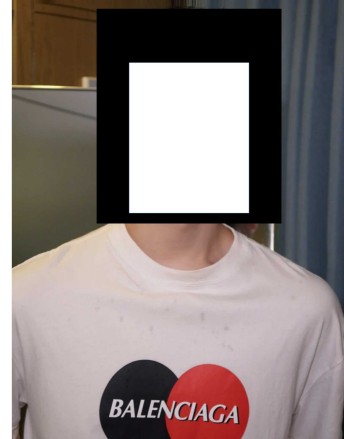

(a) Normal generation        (b) Partition generation

Figure 9: **Condition Image.** (a) Condition image of the normal inpainting model, where the black regions indicate the areas to be generated. (b) Condition image generated by hair and face partitioning, where the white regions represent the face, and the black regions correspond to the hair and background.

only verifies the effectiveness of ID preservation tasks in VAE encoding but also serves as a data construction pipeline. As shown in Figure 9, by explicitly introducing local masks, DMID-Inpainting divides the input space into facial regions and hairstyle/background regions, enabling the network to generate independently by region and optimize collaboratively. This significantly enhances the controllability and overall consistency of character generation. Additionally, DMID-Inpainting supports seamless integration with aesthetic LoRA and weight fusion with PuLID. It can flexibly construct diverse data pipelines by adjusting only a few hyperparameters. Experiments demonstrate that DMID-Inpainting exhibits strong generalization ability and ID consistency on the dataset constructed in this paper. As shown in Figure 10, DMID-Inpainting can still maintain a high degree of identity consistency in local editing tasks. Benefiting from the proposed region-wise control strategy, the disentangled representation of hairstyle and face significantly enhances the stability and controllability of each region. In practical applications, the accurate preservation of hairstyle further amplifies the benefits of identity preservation. It is worth noting that when the mask region only covers the face, DMID-Inpainting can serve as a stable face-swapping model. Additionally, using VAE as the encoder for image conditions brings consistency in details, making the edited regions more realistic. As shown in columns 3 and 5 of Figure 10, DMID-Inpainting can even accurately restore subtle facial features (such as facial moles), further verifying its advantages in detail consistency.

## A.4   High-Quality Data Construction Pipeline

Due to the use of VAE for conditional encoding, the model's requirements for the quality of data pairs are significantly higher. For pairs with low similarity, the model fails to learn consistent ID representations. Unlike facial feature encoding, where the space is relatively small, the large output space of the VAE complicates training. On the other hand, for pairs with high similarity, the model is more prone to duplication issues, resulting in a lack of facial pose diversity. This underscores the critical role of the data construction process. Unlike existing approaches that rely on millions of samples, we do not adopt a large-scale data strategy. Instead, we base our approach on the following observation: noise images and condition images share the same VAE encoding path, which inherently provides high consistency in identity features. Additionally, the diversity of poses required for face generation tasks is relatively limited. Therefore, we argue that a small amount of carefully selected high-quality data can be used to train a competitive identity-preserving model, significantly reducing training costs. To achieve this, we have designed and fully implemented a systematic data construction pipeline, including steps from raw data collection, quality evaluation, similarity filtering, to final pairing, as illustrated in the figure 11.

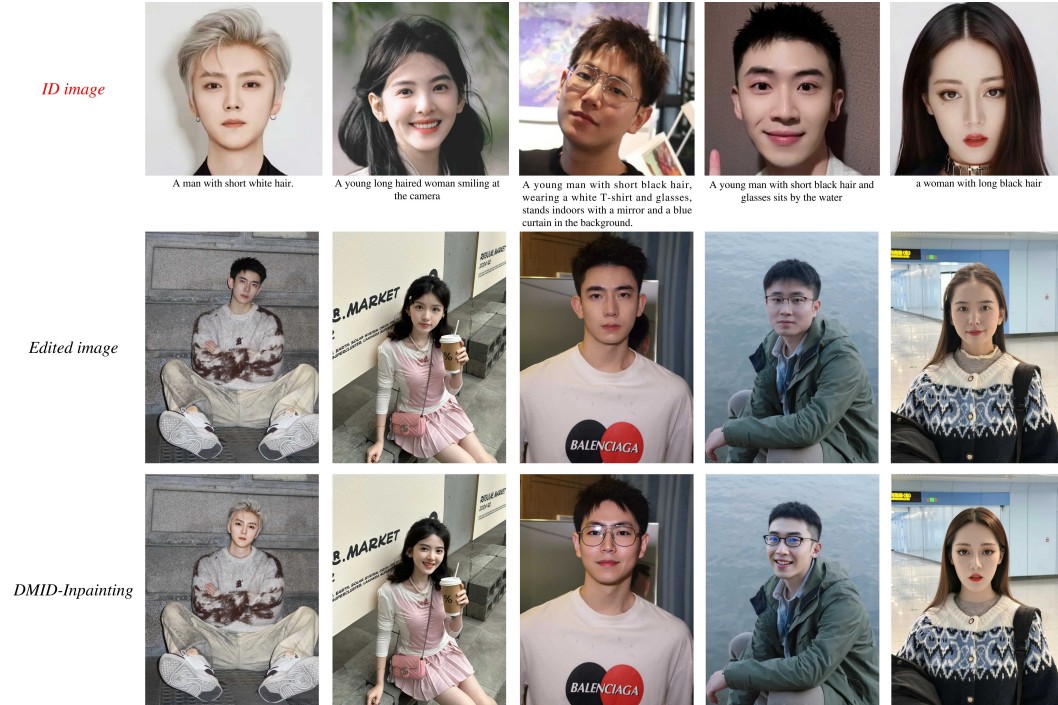

Figure 10: **DMID-Inpainting.** This figure presents the results of DMID-Inpainting, from which it can be observed that this method can stably generate high-quality data.

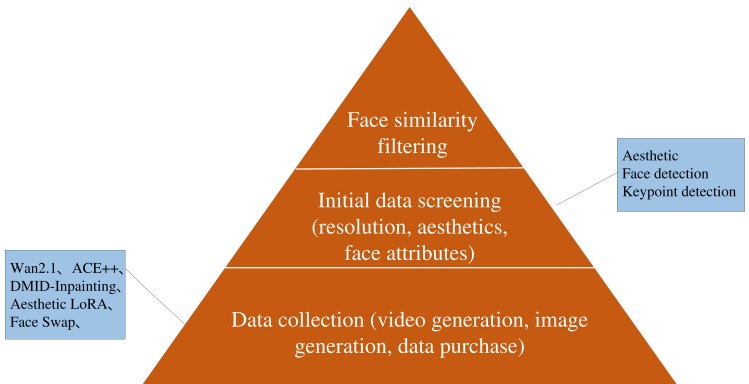

Figure 11: **Overall Data Construction Pipeline.** The overall process of data construction comprises three stages: (1) data collection, (2) data filtering, and (3) face similarity filtering.

### A.4.1 DATA COLLECTION

We utilize Wan2.1 to batch-generate videos with high-quality portraits as the initial frames. Offline frameworks such as ACE++ are employed, integrating inpainting, face swapping, and aesthetic LoRA to output high-fidelity identity-consistent images. Additionally, DMID-Inpainting is used to generate high-quality facial images. Furthermore, high-resolution portraits are collected in compliance with relevant regulations to enhance diversity.

### A.4.2 DATA FILTERING

Initial filtering is conducted using four thresholds: resolution, clarity, facial attributes (including face angle, face quality, face size, etc.), and aesthetic score. Subsequently, intra-identity pairs are retained by applying a high threshold to the facial feature similarity matrix. The entire process

involves no manual intervention, ultimately obtaining 12,000 identities with approximately 40,000 high-quality images. Experimental validation demonstrates that this dataset is sufficient to support excellent ID consistency.

### A.4.3 CONSTRUCTION OF DPO TRAINING DATA

We collected 188 ID images covering diverse ethnicities, ages, and genders. Based on these data, we constructed 300 image pairs, where one image serves as the ID image and the other as the target image. We used Qwen2-VL-7B to generate descriptions of the target images as prompts, and input the ID images (as conditional images) and these prompts into DMID-2. According to Equation 10, we selected the samples with the highest scores and the lowest scores among the 8 generation results, which were used as our training data.

