# OpenReview forum: "DMID:Dynamic Mask Attention for High-Fidelity Identity Preservation under Limited Data"
_ICLR.cc/2026/Conference — Submitted to ICLR 2026_

### Official Review · Reviewer_HcZ3 · 2025-10-30

**Soundness:** 2
**Presentation:** 2
**Contribution:** 2
**Rating:** 2
**Confidence:** 5

**Summary:**

This paper proposes DMID, a method for text-to-image (T2I) generation to balance identity consistency and textual semantics under limited data. It uses VAE for identity encoding, a lightweight AttnMaskNet with two losses (Distribution Consistency Loss, Identity Mask Loss), and a three-stage training framework. Experiments have shown the effectiveness of DMID.

**Strengths:**

1. The proposed AttnMaskNet has only about 1% of the parameters of Flux.1-dev. It generates adaptive attention masks to focus on key facial regions and suppress redundant areas, balancing identity preservation and textual semantic retention without excessive computational overhead.
2. The three-stage training sequentially optimizes identity encoding, attention distribution, and generation stability. This framework enhances model performance step-by-step, as verified by ablation experiments showing improved identity consistency and minimal semantic loss.
3. The paper establishes an ID consistency benchmark with 504 unique IDs (covering 3 age groups and 4 races), solving the problem of private, small-scale datasets in existing studies (e.g., 170 IDs in PuLID) and enabling fair comparison of different methods.

**Weaknesses:**

1. This work does not evaluate critical deployment-related metrics such as inference speed (e.g., whether the dynamic attention mask calculation increases inference latency) or resource consumption. For practical deployment scenarios with limited computing resources, the method’s feasibility and efficiency remain unvalidated.
2. Lack of comparison with SOTA models, including nano-banana, Qwen-Image.
3. Lack of user study.
4. Insufficient interpretability of the dynamic attention mask mechanism: While the paper introduces AttnMaskNet to generate dynamic attention masks and mentions visualizing attention heatmaps in ablation experiments, it lacks in-depth analysis of the mask’s decision logic.
5. Although the paper positions itself as a "limited data" solution, its training dataset still includes ~40,000 high-quality images covering 12,000 unique IDs. This weakens the persuasiveness of its "adaptability to limited data".

**Questions:**

1. This work does not evaluate critical deployment-related metrics such as inference speed (e.g., whether the dynamic attention mask calculation increases inference latency) or resource consumption. For practical deployment scenarios with limited computing resources, the method’s feasibility and efficiency remain unvalidated.
2. Lack of comparison with SOTA models, including nano-banana, Qwen-Image.
3. Lack of user study.
4. Insufficient interpretability of the dynamic attention mask mechanism: While the paper introduces AttnMaskNet to generate dynamic attention masks and mentions visualizing attention heatmaps in ablation experiments, it lacks in-depth analysis of the mask’s decision logic.
5. Although the paper positions itself as a "limited data" solution, its training dataset still includes ~40,000 high-quality images covering 12,000 unique IDs. This weakens the persuasiveness of its "adaptability to limited data".

---

### Official Review · Reviewer_tUUM · 2025-10-30

**Soundness:** 3
**Presentation:** 3
**Contribution:** 3
**Rating:** 4
**Confidence:** 3

**Summary:**

The paper presents DMID, a framework that enhances high-fidelity identity preservation under limited data through a VAE-based identity encoder, dynamic attention mask, and novel loss functions, achieving state-of-the-art identity and semantic consistency with high efficiency.

**Strengths:**

1. DMID explores the use of the reinforcement learning method Diffusion-DPO in the post-training stage to improve both identity consistency and semantic consistency.

2. Comprehensive ablation studies demonstrate the effectiveness of each module in the proposed method.

**Weaknesses:**

1. Figure 2 is too small, and the font size within the figure differs significantly from that of the caption.

2. There is a typo “generationEsser et al. (2024)” on line 047, and a typo "optimizerMishchenko & Defazio (2024)" on line 308.

3. The paper lacks comparisons with state-of-the-art methods such as FLUX.1 Kontext [1], and DreamO [2].

[1]FLUX.1 Kontext: Flow Matching for In-Context Image Generation and Editing in Latent Space

[2] DreamO: A Unified Framework for Image Customization

**Questions:**

I would be willing to increase my score if the authors adequately address my concerns.

---

### Official Review · Reviewer_pkMG · 2025-11-01

**Soundness:** 2
**Presentation:** 2
**Contribution:** 2
**Rating:** 4
**Confidence:** 2

**Summary:**

This paper introduces DMID, a method designed to achieve high identity consistency under limited-data settings. The approach integrates a dynamic attention masking mechanism with joint losses that enforce both identity fidelity and distribution consistency, and further adopts a three-stage progressive training pipeline. As a result, DMID effectively enhances identity preservation and fine-grained detail reconstruction while mitigating the inherent conflict between textual semantics and conditional image features. Extensive experiments demonstrate that DMID outperforms existing methods in both identity consistency and facial detail recovery.

**Strengths:**

- The proposed dynamic attention mask mechanism is well-motivated and technically reasonable.
- The method demonstrates strong performance in identity preservation, supported by both qualitative and quantitative results.

**Weaknesses:**

- The manuscript contains noticeable formatting problems. All in-text citations are incorrectly formatted. The authors should use “\citep” for citation rather than “\citet”. Besides, the bottom of page 2 includes a large blank space, which should be removed to meet conference formatting standards.

- Although the authors position this work as a “limited-data” approach, the training dataset includes approximately 40,000 high-quality image pairs, which is not a small-scale dataset. This undermines the claimed novelty of data efficiency.

- While the method includes a third-stage Diffusion-DPO post-training process, there is no ablation isolating the effect of this stage. The contribution of DPO to model performance remains ambiguous. In addition, terms like “DMID-1” and “DMID-2 are introduced but not clearly defined in the ablation section. A clearer explanation of these variants is needed.

**Questions:**

Please see the weaknesses.

---

### Official Review · Reviewer_eA2z · 2025-11-03

**Soundness:** 2
**Presentation:** 1
**Contribution:** 2
**Rating:** 4
**Confidence:** 5

**Summary:**

This paper proposes DMID, a diffusion-based framework for high-fidelity identity-consistent text-to-image generation. The method introduces a dynamic attention masking mechanism (AttnMaskNet) trained with two novel losses (i.e., Identity Mask Loss and Distribution Consistency Loss) to balance identity preservation and textual semantics. The approach employs a three-stage training strategy: identity embedding, dynamic mask optimization, and preference alignment via Diffusion-DPO. The authors also contribute a new ID consistency benchmark with 504 diverse identities. Experiments show that DMID outperforms existing methods like PuLID, InfiniteYou, and ACE++ in both identity similarity and text-image alignment.

**Strengths:**

1. The dynamic attention masking approach is innovative and addresses a key challenge in ID-consistent generation—balancing identity fidelity with semantic alignment.

2. The three-stage pipeline is well-structured and logically motivated, incorporating modern techniques like DPO for preference alignment.

3. The introduction of a public ID consistency benchmark with diverse demographics is a valuable contribution to the community.

**Weaknesses:**

Poor Writing and Clarity: The paper is difficult to follow due to unclear explanations, grammatical errors, and inconsistent terminology. Key concepts are often introduced without sufficient motivation or context.

Serious Formatting and Citation Issues:
  There is an issue with the citation format, missing a curly bracket.
  Figures and tables are poorly integrated and sometimes lack clear captions or references in the text.
  The overall layout is unprofessional, with inconsistent spacing, font sizes, and section numbering.

From a visual perspective (Figure 3), it can be observed that DMID does not demonstrate an advantage compared to InfiniteYou. In fact, the results in the third and fourth rows are inferior to those of InfiniteYou. This raises questions about the substantial effectiveness of the complex mechanism introduced in this paper, making the results of the study less convincing.

**Questions:**

Please refer to Weaknesses

---

### Meta-Review · Area_Chair_uZnr · 2025-12-31

**Summary:**

This paper proposes DMID, a method for identity-preserving text-to-image generation that include a dynamic attention mask mechanism and a three-stage training pipeline. The reviewers recognized the novelty of the attention masking mechanism and the value of the new benchmark dataset. However, several weaknesses were identified, particularly regarding the paper's presentation quality (formatting and citation errors, poor figure legibility), the definition of "limited data" (using 40k images contradicts the claim), and insufficient comparison with SOTA baselines like FLUX.1 Kontext and Qwen-Image. Reviewer eA2z explicitly pointed out that the visual results were inferior to InfiniteYou in some cases, questioning the effectiveness of the proposed pipeline. Reviewers pkMG and HcZ3 also criticized the lack of ablation study on the DPO post-training stage and the interpretability of the masking mechanism.

**Reviewer Concerns:**

The authors did not provide a rebuttal or response to any of the review comment.

**Reviewer Scores:**

Given the lack of a rebuttal, reviewers are expected to maintain or even lower their scores.

---

### Decision · Program_Chairs · 2026-01-26

Reject